# Optimization of Photogenerated Charge Carrier Lifetimes in ALD Grown TiO_2_ for Photonic Applications

**DOI:** 10.3390/nano10081567

**Published:** 2020-08-10

**Authors:** Ramsha Khan, Harri Ali-Löytty, Jesse Saari, Mika Valden, Antti Tukiainen, Kimmo Lahtonen, Nikolai V. Tkachenko

**Affiliations:** 1Photonic Compounds and Nanomaterials Group, Faculty of Engineering and Natural Sciences, Tampere University, P.O. Box 692, 33014 Tampere, Finland; 2Surface Science Group, Faculty of Engineering and Natural Sciences, Tampere University, P.O. Box 692, 33014 Tampere, Finland; harri.ali-loytty@tuni.fi (H.A.-L.); jesse.saari@tuni.fi (J.S.); mika.valden@tuni.fi (M.V.); 3Faculty of Engineering and Natural Sciences, Tampere University, P.O. Box 692, 33014 Tampere, Finland; antti.tukiainen@tuni.fi (A.T.); kimmo.lahtonen@tuni.fi (K.L.)

**Keywords:** titanium dioxide, atomic layer deposition, transient absorption spectroscopy, thin films, lifetime of charge carriers

## Abstract

Titanium dioxide (TiO_2_) thin films are widely employed for photocatalytic and photovoltaic applications where the long lifetime of charge carriers is a paramount requirement for the device efficiency. To ensure the long lifetime, a high temperature treatment is used which restricts the applicability of TiO_2_ in devices incorporating organic or polymer components. In this study, we exploited low temperature (100–150 °C) atomic layer deposition (ALD) of 30 nm TiO_2_ thin films from tetrakis(dimethylamido)titanium. The deposition was followed by a heat treatment in air to find the minimum temperature requirements for the film fabrication without compromising the carrier lifetime. Femto-to nanosecond transient absorption spectroscopy was used to determine the lifetimes, and grazing incidence X-ray diffraction was employed for structural analysis. The optimal result was obtained for the TiO_2_ thin films grown at 150 °C and heat-treated at as low as 300 °C. The deposited thin films were amorphous and crystallized into anatase phase upon heat treatment at 300–500 °C. The average carrier lifetime for amorphous TiO_2_ is few picoseconds but increases to >400 ps upon crystallization at 500 °C. The samples deposited at 100 °C were also crystallized as anatase but the carrier lifetime was <100 ps.

## 1. Introduction

Wide bandgap transition metal oxides (e.g., TiO_2_, ZnO, SnO_2_) have broad range of applications in photovoltaics and photocatalytic devices where they supply or deliver charge carriers [1,2]. Exploitation of these materials is crucial for advancement in many photonic applications like photovoltaics [3,4] photodegradation [5,6] and photocatalysis [7]. Metal oxides in practical applications are often polycrystalline or amorphous with high degree of lattice disorder that affects the density of band edge states [8]. These states are defect states that can act as traps and promote the deleterious recombination of charge carriers.

Among various wide bandgap transition metal oxides, TiO_2_ has been extensively employed as it shows long-term chemical stability [9] low cost and good bio-compatibility [10,11]. Photons with energy >3.2 eV generate electron-hole pairs in TiO_2_ bulk [12]. The lifetime of the photogenerated carriers is a key parameter affecting the efficiency of a photovoltaic or a photocatalytic device, and therefore, it is critical to the system design. Both, the electrons from conduction band (CB) and the holes from valence band (VB) participate in the charge transfer or directly fuel photocatalytic reactions at the TiO_2_ surface. However, the photogenerated electrons can be trapped at defect states or recombine with the holes losing their energy to the surroundings and cannot participate in the targeted actions. In order to improve the performance of TiO_2_ in photovoltaics and photocatalytic applications, it is paramount requirement that carriers do not recombine rapidly and have sufficient lifetime to diffuse through the TiO_2_ layer or to be consumed in catalytic reactions [1,13,14].

Transient absorption spectroscopy (TAS) is used to study the charge carrier dynamics ranging from femto-to millisecond timescales [15]. This time-resolved analysis provides information on the population of the photogenerated charge carriers and their kinetics in time scale relevant to photonic applications. The carrier lifetime and recombination processes depend critically on the crystal structure of TiO_2_. The long range disorder of amorphous TiO_2_ (am.-TiO_2_) results in under and over-coordinated Ti ions in which short staggered chains of edge and vertex linked Ti–O octahedral like units are present [16]. Prasai et al. [17] have proposed that long-range disorder affects the localization of band edge states. These highly localized tail states originate in am.-TiO_2_ from positional disorder of oxygen in the VB and over coordinated Ti atoms in the CB. In addition, due to low vacancy formation energy in am.-TiO_2_, it has relatively abundant oxygen vacancies as compared to its crystalline counterparts [8,18] which creates more disorder and hence defect states. These defect states can trap electrons and holes which leads to increased density of recombination states [19]. Therefore, the heat treatment is usually applied to convert am.-TiO_2_ to crystalline TiO_2_ which reduces disorder and increases the lifetime of the photocarriers.

Tremendous efforts have been made to employ different fabrication routes for preparing various TiO_2_ nanostructures such as thin films, nanoparticles, nanorods and hierarchical structures to increase their performance in photonic applications. Among various methods atomic layer deposition (ALD) has gained increasing interest as it provides high quality thin films with controlled thickness [20], conformal coverage and high reproducibility [7]. For TiO_2_ growth by ALD, TiCl_4_ and H_2_O are the most commonly used precursors, but the process requires relatively high temperature and forms HCl as a by-product which is corrosive to the ALD equipment. Instead, growth using tetrakis(dimethylamido)titanium(IV) (TDMAT; metal-amide compound) has become more popular since it is more reactive than TiCl_4_ (metal-halide compound) allowing growth at lower temperature and does not form corrosive by-products with H_2_O [21,22]. The growth temperature affects the amount of precursor traces in ALD TiO_2_ thin films [23] which can have strong effects on crystallization and charge carriers kinetics [24]. Therefore, for photonic applications requiring long charge carrier lifetimes the optimization of both the growth temperature and the temperature of subsequent heat-treatment should be considered.

In the present work, we have employed ALD technique at low temperature (100–150 °C) to prepare TiO_2_ thin films from TDMAT and H_2_O. TDMAT has been used for this study as this precursor allows TiO_2_ deposition at low temperature. Low deposition temperature was exploited to study a possibility of depositing TiO_2_ thin films on thermally subtle materials such as organics and polymers compounds of organic solar cells such as PEDOT which decomposes at temperature above 390 °C [22,25,26]. The aim of this study was to minimize the deposition and heat treatment temperatures without compromising the lifetime of photogenerated charge carriers. The comparison was made between the TiO_2_ films grown at 100 °C and 150 °C, which both crystallizes as anatase upon annealing in air. A higher deposition temperature of 200 °C was studied previously [18] and it was shown to crystallize as rutile phase upon the same heat treatment. TAS was done on ultrafast time scales in air environment to study the kinetics of photogenerated charge carriers within TiO_2_. It was observed that heat treatment enhanced the lifetime of charge carriers up to 100 folds as compared to the as-deposited am.-TiO_2_ samples, but the enhancement depended on the ALD growth temperature. Hence, TiO_2_ thin films with long lifetime of charge carriers can be fabricated at low temperature heat treatment when ALD growth temperature is first optimized.

## 2. Materials and Methods

### 2.1. Substrates

UV-grade fused silica (quartz) (10 × 10 × 1 mm) from Präzisions Glas & Optik GmbH (Iserlohn, Germany) was used as a substrate in optical measurements. In addition, degenerately Sb-doped (resistivity 0.008–0.02 Ω·cm) n-type Si(100) wafers from Siegert Wafer GmbH (Aachen, Germany) cleaved in 10 × 10 × 0.525 mm pieces were used as substrates in GIXRD and SEM experiments.

### 2.2. Synthesis

ALD deposition of TiO_2_ was carried out using a Picosun Sunale ALD R200 Advanced reactor. Tetrakis(dimethylamido)titanium(IV) (Ti(N(CH_3_)_2_)_4_, TDMAT, electronic grade 99.999+%, (Sigma-Aldrich, Inc., Taufkirchen, Germany), ultrapure Milli-Q water, and Ar (99.9999%, Oy AGA Ab, Tampere, Finland) were used as the Ti precursor, O precursor, and carrier/purge/venting gas, respectively. During the ALD, the substrate temperature was kept at 100 or 150 °C, respectively. The film growth rate was calibrated by ellipsometry (Rudolph Auto EL III Ellipsometer, Rudolph Research Analytical, Hackettstown, NJ, USA). The TiO_2_ film thickness of 30 nm (480 and 636 ALD cycles at 100 °C and 150 °C, respectively) was used for all the experiments. The precursors for ALD TiO_2_ and the 150 °C growth temperature were chosen based on the work by Hu et al. [27]. The post heat treatment in air was carried out by placing the sample into a pre-heated tube furnace for 45 min.

### 2.3. Characterizations

The phase structure of the samples was defined via Grazing Incidence X-ray Diffraction (GIXRD, Panalytical X′ Pert PRO and Panalytical Empyrean diffractometers, Malvern Panalytical Ltd., Malvern, UK) with Cu Kα radiation (λ = 1.5405 Å, hv = 8.04 keV) and 45 kV and 40 mA cathode voltage and current, respectively. The samples were scanned in 2θ between 24 and 34° by using grazing-incidence angle of 0.3° for X-rays. The surface morphology of as-deposited and heat treated TiO_2_ thin films on Si substrate was studied by scanning electron microscopy (SEM, Zeiss Ultra 55, Carl Zeiss Microscopy GmbH, Oberkochen, Germany).

Transient absorption spectra were measured using pump-probe setup in sub-ps to ns timescales. Both transmittance and reflectance modes were used to study the samples. The fundamental laser pulses were generated using Ti:Sapphire laser (Libra F, Coherent Inc., 800 nm, ~100 fs pulse width at repetition rate of 1 kHz). The main part of the beam was directed on to the optical parametric amplifier (Topas C, Light Conversion Ltd., Vilnius, Lithuania) to produce the desired wavelength (320 nm in our case with an energy density of 100 µJ cm^−2^). The rest of the fundamental beam was directed to time resolved spectrophotometer (ExciPro, CDP Inc.) where it was passed through a delay line and focused on a sapphire to generate white continuum used as the probe. The probe light was further split into reference and signal beams which were focused over the samples. TiO_2_ samples deposited on quartz substrates were slightly tilted in respect to the incident probe so that both transmitted and reflected beams could be measured in identical conditions. Details of the measurements and data analysis were published by Pasanen et al. [28].

## 3. Results and Discussion

Figure 1 shows steady state UV-Vis spectroscopy analysis for the sample series deposited at 150 °C and heat treated at different temperatures. The absorbance spectra (A) were calculated from transmission (T, Figure 1a) and reflectance (R, Figure 1b) measurements as A = −log[T/(1 − R)]. The optical bandgaps are determined from Tauc plots as shown in Figure 1d, which show only little variation and were 3.4 ± 0.1 eV for all the samples. The most notable change in the optical absorption took place between the samples heat treated at 250 °C and 300 °C. Up to 250 °C heat treatment, the absorbance increased smoothly at the wavelengths shorter than that corresponding to the bandgap edge as shown in Figure 1c. In contrast, for temperatures >250 °C, the results were virtually similar with each other showing sharper absorption edge and a clear change in the absorption slope at 3.9 eV.

The high degree of band edge tailing states typical for amorphous TiO_2_ result in a lower slope of absorption rise toward UV region for the as-deposited, 200 °C and 250 °C heat treated samples as compared to the samples heat treated at higher temperatures. The sharper absorption edge with a distinct change in the slope indicated in the Tauc plot at 3.9 eV, Figure 1d for samples heat treated at temperature higher than 250 °C is a signature of anatase TiO_2_ thin film with high degree of crystalline order [17,29,30]. Interestingly, the tailing of electronic states into the bandgap had little effect on the optical bandgap, which is in accordance with the theoretical work by Prasai et al. [17] where they found that optical properties of am.-TiO_2_ are similar to anatase TiO_2_ despite the highly localized tail states predicted for am.-TiO_2_. The UV-Vis data for the ALD TiO_2_ thin films grown at 100 °C (Appendix A) were similar with the 150 °C growth temperature, except for the temperature where the change in the absorption edge was observed, between 300 °C and 400 °C as shown in Appendix A. It has been shown previously, that further increase of ALD growth temperature from 150 °C to 200 °C results in a significant increase in absorption in the visual range due to the formation of visually black and electrically leaky am.-TiO_2_ [18].

Figure 2 shows grazing-incidence X-ray diffraction (GIXRD) patterns for ALD TiO_2_ thin films grown at 100 °C and 150 °C and heat treated in air. Results show that TiO_2_ thin films deposited by ALD at 100 °C and 150 °C were both amorphous. However, when the heat treatment temperature was increased, the crystallization of am.-TiO_2_ to anatase phase was identified based on the most intensive XRD peak of at 25.6° [31] which was clearly seen after heat treatment at 300 °C and higher for the 150 °C growth temperature. The samples deposited at 100 °C required higher temperature ≥375 °C heat treatment to crystallize into anatase. This confirms the assignment of the change observed in the absorption edge to the crystallization of am.-TiO_2_.

SEM images of amorphous and at 400 °C heat-treated samples are depicted as insets in Figure 2 which reveal another distinct difference between the two growth temperatures. Growth temperature mediates the morphology of anatase TiO_2_ grains upon crystallization. The 30 nm thick thin film grown at 100 °C depicts some exceptionally large (~30 µm) grains after crystallization. In strict contrast, thin films grown at 150 °C and heat-treated at 400 °C resulted in more homogenous and uniformly distributed TiO_2_ grains with no such large grains present as in the case of 100 °C deposited heat-treated samples. Since grain morphology affects defect states, a difference in the carrier lifetime is expected for the two growth temperatures [32]. The explosive crystallization of ALD grown amorphous TiO_2_ into anatase TiO_2_ has been previously reported to result from Nb_2_O_5_ or Ta_2_O_5_ doping [33]. We assign the difference in crystallization preliminary to the temperature dependence of TDMAT precursor traces in the as-deposited ALD TiO_2_ thin films. We have previously shown that the crystallization of similar ALD TiO_2_ thin film grown at 200 °C is accompanied by surface segregation of N species [18]. Nitrogen in am.-TiO_2_ increases nucleation temperature [34] and inhibits anatase to rutile phase transition [24].

Samples were studied by TA spectroscopy to investigate the charge carrier dynamics from a picosecond to nanosecond timescale. Ultrashort laser pulses at 320 nm wavelength were used to excite the samples under ambient air conditions. The TA spectra were measure in the near-infrared regions (840–1020 nm). TA measurements of samples deposited at 150 °C and heat treated at 200 °C and 500 °C were carried out in transmittance and reflectance modes. The spectra at a few preselected delay times are presented in Appendix A. The intensities of transmitted and reflected signals, ΔO.D., were roughly the same for the sample heat-treated at 200 °C (am.-TiO_2_). However, for the 500 °C heat-treated sample (anatase TiO_2_), the signal obtained in reflectance mode, ΔR, was stronger than that in transmittance mode, ΔT, under otherwise the same conditions. This difference in TA responses of low and high temperature treated samples can be tentatively attributed to the difference of the sample crystallinity, and thus, different relative change of refractive index and absorption coefficient [28] occurring after excitation as shown in Appendix A. Decays were fitted by single stretched-exponent in delay times longer than 0.4 ps. The fast (<0.4 ps) response was not strong and is most probably due to thermal relaxation of hot carriers. Details of fitting are presented in the SI.

The essential feature of the TA decay is that it is virtually wavelength independent in the IR range and the same fit results were obtained at different wavelengths and out of a global fit. Therefore, to compare decay profiles, one can select a single wavelength for all different samples. Figure 3a,b shows normalized TA decay profiles of samples measured in reflectance mode at 960 nm for 100 and 150 °C deposited and heat-treated samples, respectively. Change in optical density, ΔOD, is proportional to the number of charge carries generated by the excitation. The decay profiles show that the carrier lifetime increases tremendously as the heat treatment temperature increases from 250 to 300 °C for samples grown at 150 °C, and after that the lifetime is virtually independent of heat treatment temperature. It can be concluded that the heat treatment induced crystallization of am.-TiO_2_ into anatase TiO_2_ at 300 °C for samples grown at 150 °C is the main reason for the abrupt increase of the carrier lifetimes. Further increase of the heat treatment temperature to 500 °C has only minor effect on the lifetime which can be attributed to removal of remaining defect sites in the TiO_2_ film.

The decay profiles for the same heat-treatment series on the growth temperature of 100 °C is shown in Figure 3a and evidence the same trend with the 150 °C grown samples as shown in Figure 3b. The lifetime of heat-treated samples increases gradually in both cases, but the sample deposited at 150 °C show much larger lifetime increase, and the change is sharper–most of the rise takes place when the heat treatment temperature was increased from 250 °C to 300 °C as shown in Figure 3b. Further increase of the deposition temperature results in rather drastic changes in the sample structure as has been shown previously [18].

ALD deposition at 200 °C yields a new type of film, so-called black titania which cannot be directly compared with optically transparent am.-TiO_2_ presented here. The heat treatments of such film leads to rutile crystal structure rather than anatase TiO_2_. This makes comparison of samples deposited at 150 °C and 200 °C controversial. However, if the only parameter of interest is the carrier lifetime, heat-treated samples deposited at 200 °C (rutile TiO_2_) had similar lifetime as compared to the samples deposited at 150 °C (anatase TiO_2_), as shown in Appendix A.

The dependence of carrier lifetime on heat treatment temperature for both 100 °C and 150 °C deposited sample series is presented in Figure 3c. As the heat treatment temperature is increased, the lifetime increases sharply for samples heat treated at 250 °C deposited at 150 °C as compared to 350 °C heat treated samples deposited at 100 °C. Also, one can notice a minor increase in absorbance at 320 nm in the steady state spectra as shown in Figure 1c and Appendix A following the increase in heat treatment temperature for both series of samples deposited at 100 °C and 150 °C. The key changes in the absorbance and lifetimes of TiO_2_ thin films deposited at 100 °C and 150 °C take place in the temperature range of 350–400 °C and 250–300 °C, respectively as shown in Figure 4. Apparently, one plausible reason for these changes is the transition from amorphous to polycrystalline anatase form. This is also confirmed by GIXRD data shown in Figure 2.

## 4. Conclusions

In summary, ALD TiO_2_ samples grown at 100 °C and 150 °C have amorphous structure and fast recombination (2–10 ps) of charge carriers presumably due to high density of tail states. As the post heat treatment temperature is increased over 250 °C, amorphous structure is converted into anatase. This conversion from amorphous to crystalline anatase phase results in more than two orders of magnitude increase in carriers lifetime (i.e., >400 ps). As the longer lifetime of carriers is important for efficient performance for photoelectrodes, the best results at the lowest possible temperature were obtained for samples deposited at 150 °C and heat-treated at 300 °C. Further increase in the heat treatment temperature gives only minor gain to the carrier lifetime and is restrictive, especially when the photonic system includes organic or polymer components.

## Figures and Tables

**Figure 1 nanomaterials-10-01567-f001:**
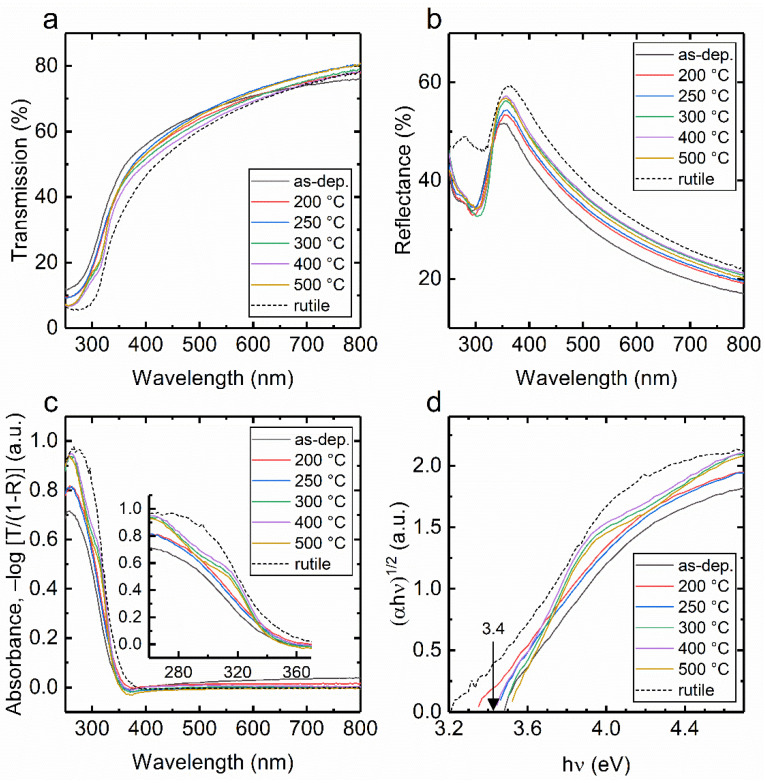
(**a**) Transmittance spectra, (**b**) reflectance spectra, (**c**) absorbance spectra and (**d**) Tauc plots of the ALD TiO_2_ thin films grown at 150 °C and heat treated at different temperatures indicated in the plots. The spectra of a rutile thin film of equal thickness is show as dotted lines using the data acquired from Ref. [18].

**Figure 2 nanomaterials-10-01567-f002:**
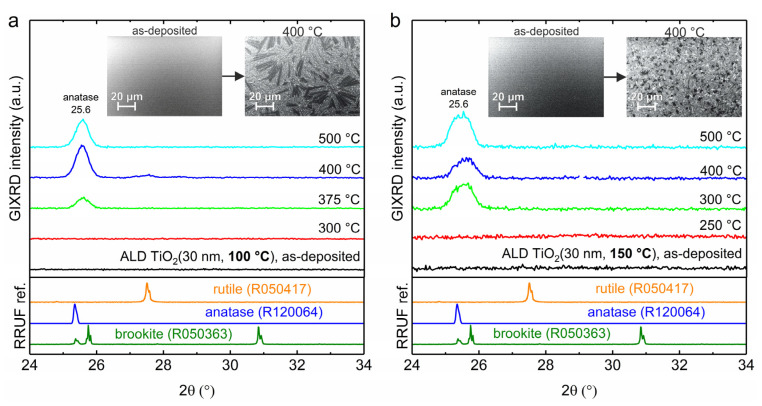
GIXRD patterns for ALD TiO_2_ thin films grown at (**a**) 100 °C and (**b**) 150 °C and heat treated in air. The insets show SEM images for the as-deposited and at 400 °C heat-treated samples. XRD reference patterns for rutile, anatase and brookite TiO_2_ are from RRUFF database [31].

**Figure 3 nanomaterials-10-01567-f003:**
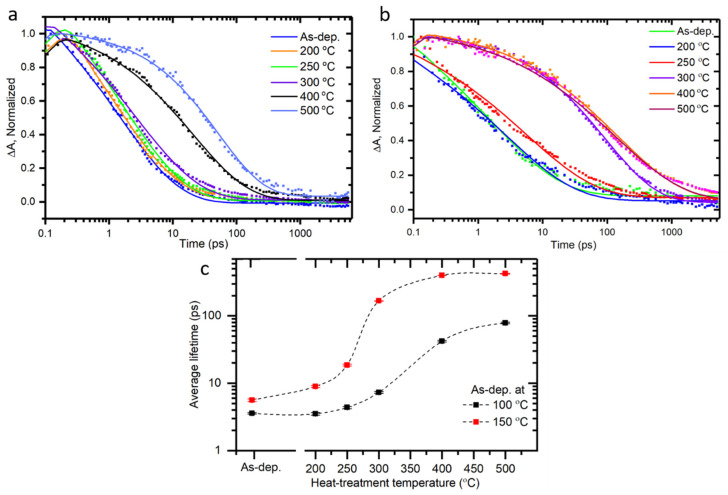
Normalized transient absorption decays profile at 960 nm of samples deposited at (**a**) 100 and (**b**) 150 °C without heat treatment temperature indicated in the plot (**c**) Average lifetimes as function of the heat-treatment temperature for samples deposited at 100 and 150 °C.

**Figure 4 nanomaterials-10-01567-f004:**
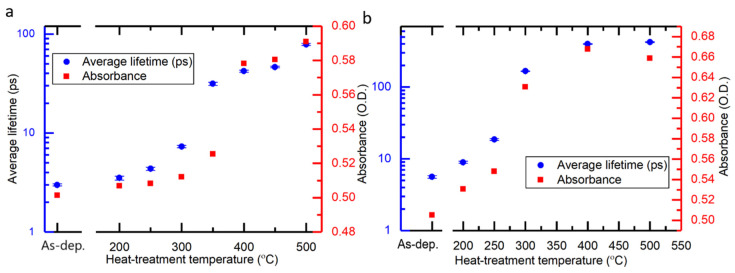
Average lifetime as a function of heat-treatment temperature shown along with absorbance of the samples at 320 nm from steady state spectra for samples grown at (**a**) 100 °C and (**b**) 150 °C.

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
