# Peer review of "Optimization of Photogenerated Charge Carrier Lifetimes in ALD Grown TiO2 for Photonic Applications"

_nanomaterials, 2020, doi:10.3390/nano10081567_

Round 1

Reviewer 1 Report

This manuscript presents the deposition of TiO2 thin films at low temperature via ALD and TDMAT precursor for a long lifetime of charge carriers. The authors demonstrated the optimal result from the TiO2 films grown at 150 degree C and heat-treated at as low as 300 degree C. The results will be informative for photocatalytic and photovoltaic applications of TiO2 thin films, but the manuscript lacks the description of the novelty and the mechanisms of the TiO2 film growth.

  • In the introduction part, the authors describe the importance of converting amorphous into crystalline TiO2 to reduce disorder or defect states to increase the lifetime of the photocarriers. However, the originality of the manuscript is unclearly addressed in the introduction. It appears that lower temperature ALD and the use of TDMAT might be new in comparison with literature, which is considered as an incremental contribution.
  • Figure 1 results present the promoted crystallization of TiO2 when higher deposition temperature was used. However, the manuscript does not include a discussion to explain the role of deposition temperature.
  • Figure 2 XRD results also confirm the conversion of anatase at a lower temperature when 150 degree C is used for TiO2 deposition in comparison with 100 degree C. Again, the authors did not address an explanation about conversion and morphology development driven by deposition temperature.

Author Response

Point 1: This manuscript presents the deposition of TiO2 thin films at low temperature via ALD and TDMAT precursor for a long lifetime of charge carriers. The authors demonstrated the optimal result from the TiO2 films grown at 150 degree C and heat-treated at as low as 300 degree C. The results will be informative for photocatalytic and photovoltaic applications of TiO2 thin films, but the manuscript lacks the description of the novelty and the mechanisms of the TiO2 film growth.

Response 1: We thank the reviewer for his/her positive view on our TiO2 optimisation protocol and its importance for practical application, and we appreciate reviewer’s critical points on the manuscript and aimed our revision on addressing the issues, namely adding description and emphasizing novelty as requested.

Point 2: In the introduction part, the authors describe the importance of converting amorphous into crystalline TiO2 to reduce disorder or defect states to increase the lifetime of the photocarriers. However, the originality of the manuscript is unclearly addressed in the introduction. It appears that lower temperature ALD and the use of TDMAT might be new in comparison with literature, which is considered as an incremental contribution.

Response 2: TDMAT has been used previously and it was selected in this study because this precursor allows low ALD temperature for TiO2. We have added explanation and new references in  row 88-93, “The growth temperature affects the amount of precursor traces in ALD TiO2 thin films [23] which can have strong effects on crystallization and charge carriers kinetics [24]. Therefore, for photonic applications requiring long charge carrier lifetimes the optimization of both the growth temperature and the temperature of subsequent heat-treatment should be considered.” Also, at row 95 we have added “TDMAT has been used for this study as this precursor allows TiO2 deposition at low temperature.”

Point 3: Figure 1 results present the promoted crystallization of TiO2 when higher deposition temperature was used. However, the manuscript does not include a discussion to explain the role of deposition temperature.

Response 3: Please see response to the following point.

Point 4: Figure 2 XRD results also confirm the conversion of anatase at a lower temperature when 150 degree C is used for TiO2 deposition in comparison with 100 degree C. Again, the authors did not address an explanation about conversion and morphology development driven by deposition temperature. 

Response 4: The growth temperature affects the amount of N from precursor (TDMAT) of TiO2 in ALD thin films [23] which has strong effects on crystallization [24], as mentioned now in the new version of Introduction. Following recommendations of the reviewer we have added a reference to N effect on the crystallization process (rows 219-220):  “Nitrogen in am.-TiO2 increases nucleation temperature [34] and inhibits anatase to rutile phase transition [24]”

Reviewer 2 Report

Regarding the novelty and quality of science and the presentation, the submitted draft deserves to be accepted after minor revision.

Due to their importance figures, S3 and S4 from the SI should be transferred in the main text and discusses. The authors should try to make a panel figure regarding measured decay.
Second, the observed hysteresis( average lifetime as a function of temperature, Figure 3b in the main text) should be discussed more in-depth.

Author Response

Regarding the novelty and quality of science and the presentation, the submitted draft deserves to be accepted after minor revision.

Point 1: Due to their importance figures, S3 and S4 from the SI should be transferred in the main text and discusses. The authors should try to make a panel figure regarding measured decay.

Response 1: We appreciate the reviewer positive evaluation of our work and his/her suggestions. We have moved Figs. S3 and S4 to the main text. Figure S3 is merged in Figure 3 and is as Figure 3a in the main text. The caption of Figure 3 has been updated in row 240-242. Text in rows 247-249 have been edited. Figure S4 has been shifted from Supplementaly information and is Figure 4 in the main text. All figure numbers from row 264 to 287 have been updated.

Point 2: Second, the observed hysteresis( average lifetime as a function of temperature, Figure 3b in the main text) should be discussed more in-depth.

Response 2: Discussion of Figure 3b which is now Figure 3c has been added from row 279 to 282: “As the heat treatment temperature is increased, the lifetime increases sharply for samples heat treated at 250 °C deposited at 150 °C as compared to 350 °C heat treated samples deposited at 100 °C.”

Reviewer 3 Report

This paper focuses on the growth of TiO2 thin films by means of the ALD technique, in order to optimize the process parameters to gain low temperature deposition and good lifetime of photogenerated carriers. The authors also paid attention to post deposition processes, mandatory to improve the crystallinity of the material: the goal is to reach the anatase crystalline form with the lowest annealing temperature.

The paper, though not extremely innovative, is clear and contains the right amount of description related to the experimental part and the discussion.

The title, however, does not include any word that guides towards the aim of the work, so I suggest to modify it in order to let the title more focused.

The introduction, in general, provides enough references but when describing the TAS technique a general reference is missing. I suggest to include it at the end of the paragraph started at row 60.

About the post-processing temperatures, the authors claim that lowering the T helps in moving towards an ALD applications onto organic and polymeric substrates. This statement is somehow general and I suggest to include a more precise argumentation, comparing the process temperature with the feasibility the of the growth on these substrates, taking into account the lifetime of the carriers in those experimental conditions.

More detailed points to be fixed are listed here:

Figure 1)  the colours of the plots are too similar, it is difficult to distinguish the curves for the different annealing temperatures, they should be recoloured in a different way.

Sentence started at row 187) It is not clear and confused, I suggest to re-write it.

English text to be corrected: row44 (affects); row 76 (increases);row 178 (were, instead of was); row 256 (takes place)

Author Response

This paper focuses on the growth of TiO2 thin films by means of the ALD technique, in order to optimize the process parameters to gain low temperature deposition and good lifetime of photogenerated carriers. The authors also paid attention to post deposition processes, mandatory to improve the crystallinity of the material: the goal is to reach the anatase crystalline form with the lowest annealing temperature.

Point 1: The paper, though not extremely innovative, is clear and contains the right amount of description related to the experimental part and the discussion. The title, however, does not include any word that guides towards the aim of the work, so I suggest to modify it in order to let the title more focused.

Response 1: We thank reviewer for the comments and suggestions, and agree that the title can be changed to reflect on the paper content better. We have changed title to “Optimization of photogenerated charge carrier lifetimes in ALD grown TiO2 for photonic applications”.

Point 2: The introduction, in general, provides enough references but when describing the TAS technique a general reference is missing. I suggest to include it at the end of the paragraph started at row 60.

Response 2: Reference [15] has been added at row 61.

Point 3: About the post-processing temperatures, the authors claim that lowering the T helps in moving towards an ALD applications onto organic and polymeric substrates. This statement is somehow general and I suggest to include a more precise argumentation, comparing the process temperature with the feasibility the of the growth on these substrates, taking into account the lifetime of the carriers in those experimental conditions.

Response 3: References [25] and [26] have been added in row 99-100. The text has been modified as:  “Low deposition temperature was exploited to study a possibility of  depositing TiO2 thin films on thermally subtle materials such as organics and polymers compounds of organic solar cells such as PEDOT which decomposes at temperature above 390 °C [22,25,26].”

More detailed points to be fixed are listed here:

Point 4: Figure 1)  the colours of the plots are too similar, it is difficult to distinguish the curves for the different annealing temperatures, they should be recoloured in a different way.

Response 4: The colours of the plots are recoloured.

Point 5: Sentence started at row 187) It is not clear and confused, I suggest to re-write it.

Response 5: The sentences have been rewritten as “Results show that TiO2 thin films deposited by ALD at 100 °C and 150 °C were both amorphous. However, when the heat treatment temperature was increased, the crystallization of am.-TiO2 to anatase phase was identified based on the most intensive XRD peak of at 25.6° [31] which was clearly seen after heat treatment at 300 °C and higher for the 150 °C growth temperature.” from line 191 to 196.

Point 6: English text to be corrected: row44 (affects); row 76 (increases);row 178 (were, instead of was); row 256 (takes place)

Response 6: All grammatical errors have been corrected.

Round 2

Reviewer 1 Report

The authors had addressed the comments raised by this reviewer. The manuscript appears to be good for the acceptance.